

# A large set of potential past, present and future hydro-meteorological time series for the UK

Benoit P. Guillod[1], Richard G. Jones[1,2], Simon J. Dadson[3], Gemma Coxon[4], Gianbattista Bussi[1], James Freer[4], Alison L. Kay[5], Neil R. Massey[1], Sarah N. Sparrow[6], David C. H. Wallom[6], Myles R. Allen[1], and Jim W. Hall[1]

[1]Environmental Change Institute, University of Oxford, Oxford, United Kingdom
[2]Met Office Hadley Centre, Exeter, United Kingdom
[3]School of Geography and the Environment, University of Oxford, South Parks Road, Oxford OX1 3QY, United Kingdom
[4]Geographical Sciences, University of Bristol, Bristol, United Kingdom
[5]Centre for Ecology and Hydrology, Wallingford, United Kingdom
[6]Oxford e-Research Centre, University of Oxford, Oxford, United Kingdom

*Correspondence to:* Benoit Guillod (benoit.guillod@ouce.ox.ac.uk)

**Abstract.** Hydro-meteorological extremes such as drought and heavy precipitation can have large impacts on society and the economy. With potentially increasing risks associated with such events due to climate change, properly assessing the associated impacts and uncertainties is critical for adequate adaptation. However, the application of risk-based approaches often requires large sets of extreme events, which are not commonly available. Here, we present such a large set of hydro-meteorological

5  time series for recent past and future conditions for the United Kingdom based on weather@home2, a modelling framework consisting of a global climate model driven by observed or projected sea surface temperature and sea ice which is downscaled to 25km over the European domain by a regional climate model. Sets of 100 time series are generated for each of (*i*) a historical baseline (1900–2006), (*ii*) five near future scenarios (2020–2049) and (*ii*) five far future scenarios (2070–2099). The five scenarios in each future time slice all follow the Representative Concentration Pathway 8.5 (RCP8.5) and sample

10  the range of sea surface temperature and sea ice changes from CMIP5 models. Validation of the historical baseline highlights good performance for temperature and potential evaporation, but substantial seasonal biases in mean precipitation, which are corrected using a linear approach. For extremes in low precipitation over a long accumulation period (>3 months) and shorter duration high precipitation (1–30 days), the time series generally represents past statistics well. Future projections show small precipitation increases in winter but large decreases in summer on average, leading to an overall drying, consistently with the

15  most recent UK climate projections (UKCP09) but larger in magnitude than the latter. Both drought and high precipitation events are projected to increase in frequency and intensity in most regions, highlighting the need for appropriate adaptation measures. Overall, the presented dataset is a useful tool for assessing the risk associated with drought and more generally with hydro-meteorological extremes in the UK.



# 1 Introduction

Extreme weather events such as droughts can have huge socio-economic consequences, so ensuring that society is well prepared to face such events will have multiple benefits. Anthropogenic climate change is expected to have an impact on extreme events: warm temperature extremes and heavy precipitation extremes have been shown to have increased due to human greenhouse gas emissions and these trends are projected to increase in the future (IPCC, 2013). These changes will increase risks in many regions and adequate adaptation will be critical to limit the associated damages.

Despite clear trends and predicted increases in these extremes, understanding the implications for more complex hydrometeorological extremes remains limited. This is the case of drought (e.g., Sheffield et al., 2012), for which the attribution of observed (projected) trends can only be done with low (medium) confidence (IPCC, 2013) due, among other factors, to observational uncertainty and confounding effects from decadal-scale variability combined with relatively small samples due to the comparatively long duration of droughts versus other extreme events. Nonetheless, some highlighted regions may be expected to experience more frequent or more intense droughts due to climate change (the Mediterranean region, Central North America, Central America and Mexico, Northeast Brazil and southern Africa; Seneviratne et al., 2012). Another complication is that drought can be caused by various factors including precipitation deficit, excessive potential evapotranspiration (due to enhanced radiation, wind speed or water pressure deficit), and pre-conditioning (pre-event land water storage including soil moisture, snow, lake and/or groundwater storage) (Seneviratne et al., 2012). Moreover, it can be defined in multiple ways as negative anomalies in precipitation ("meteorological drought"), soil moisture ("agricultural drought") or streamflow, lake or groundwater levels ("hydrological drought").

In the United Kingdom (UK), the issue of drought or, more generally, water scarcity, has been highlighted during the 2010–2012 drought. This drought drew attention to the potentially high economic losses that would result from a severe water restriction and prompted recognition that changes in climate and in water demand may increase the risk of such an event in the future highlighted the need to better assess the risk associated with drought in the UK. The MaRIUS project (Managing the Risks, Impacts and Uncertainties of drought and water Scarcity, http://www.mariusdroughtproject.org) thus aims at better understanding the physical mechanisms and the inter-sectoral interactions leading to water scarcity, in order to support a risk-based approach for drought management.

Given the long duration, spatial variability and multi-variate nature of droughts, large sets of potential drought events are required, in order to assess the impacts of these on various sectors and to apply a risk-based approach. Available data such as the most recent set of UK Climate Projections (UKCP09, Murphy et al., 2009) provide a large range of possible climate change signals, as well as long time series at any location derived from a weather generator. However, these long time series are not spatially consistent, i.e., one cannot examine interactions between spatially distributed locations, which is critical to the mentioned risk-based assessment, particularly for drought. Other potential sources of data include climate model output from the Fifth coupled Climate Model Intercomparison Projection (CMIP5, Taylor et al., 2012); however, while these provide a wide sampling of modelling uncertainty, they do so with a limited number of transient simulations for each model. The implied low



number of simulated years impedes a proper estimation of the risk associated with rare events and, therefore, the application of risk-based management approaches, for which a large number of spatially-consistent drought events are required.

Therefore, a new set of climate time series is created using weather@home2 (Guillod et al., 2016), an improved atmosphere-only global climate model which is dynamically downscaled over a limited domain by a regional climate model (RCM) and run
on volunteers' computers around the globe. Hydro-meteorological variables for hundreds of time series are generated over the UK for the recent past and for future time slices (RCP8.5 with climate response uncertainties), from which drought events can be identified. However, the use of the time series is not restricted to drought studies but can be applied to any type of extreme event. With about 3000 years of data for each 30-year period and scenario, the created dataset allows to examine the very rare (and most severe) events with a high statistical confidence, albeit with limitations associated with the use of model-based data.
This paper presents the new hydro-meteorological climate time series. Section 2 describes the weather@home 2 model as well as observational datasets used for validation of the time series. The model simulations and the generation of the time series are detailed in Sect. 3. The time series covering the recent past are validated in Sect. 4, while the main features of the future projections are shown in Sect. 5.

## 2 Model and Data

### 2.1 Weather@home 2

Weather@home (Massey et al., 2015) consists of an atmospheric global climate model (GCM), HadAM3P, and its regional counterpart, the regional climate model (RCM) HadRM3P, which dynamically downscales the GCM to a higher resolution over a limited domain. As part of the *climateprediction.net* project (Allen, 1999), weather@home takes advantage of computing time donated by volunteers around the world to run very large numbers of climate model simulations, of the order of tens of
thousands.

The data analysed in this study is based on version 2 of weather@home (hereafter, w@h2, see Guillod et al., 2016), which uses the more recent land surface scheme MOSES 2. The regional model covers the European CORDEX domain at a horizontal resolution of $0.22°$ (about $25 \times 25$km) on a rotated longitude/latitude grid (e.g., Kotlarski et al., 2014). The model, including its setup for this domain, is described and validated in detail by Guillod et al. (2016).

### 2.2 Observational data

The gridded datasets listed in Table 1 are used for comparison and validation. For temperature, the E-OBS dataset (Haylock et al., 2008) is selected, as it is conveniently available on the same rotated longitude-latitude grid as HadRM3P. For precipitation, we use the CEH-GEAR dataset (Keller et al., 2015), which provides rainfall on a 1 km grid from 1890–2015. Observational estimates of potential evaporation are taken from the CHESS dataset (Robinson et al., 2015), available from
1961–2012 and derived with two formulations, with and without correction for interception evaporation. For both CEH-GEAR and CHESS, data are aggregated onto the $0.22°$ model grid prior to all analyses.





## 3 Methodology

### 3.1 Model simulations

A total of 11 large ensembles ("batches") of w@h2 simulations are conducted, producing model output for three distinct time periods and a range of scenarios (see Table 2). The three time periods cover the past century ("historical baseline"; 1900–2006), and two 30-year future time slices (near and far future; 2020–2049 and 2070–2099, respectively) assuming the high greenhouse gas emission scenario RCP 8.5 (Meinshausen et al., 2011). For each future time slice, uncertainty in transient climate response is taken into account by sampling a range of five sea surface temperature (SST) warming patterns derived from the fifth phase of the Coupled Model Intercomparison Project (CMIP5, Taylor et al., 2012), as detailed in Sect. 3.2, while for the historical baseline only one ensemble is generated, using the observed ocean state, leading to the total of 11 batches (1 batch for each time period and SST pattern).

All ensembles are generated using the same overarching design, described in Guillod et al. (2016) for the historical baseline. Essentially, simulations are initialised on the 1st of December before each simulated year (e.g., 1st of December from 2019–2048 for near future), using restart files from earlier 12-month spin-up simulations, and are run for 13 months. The aim is to produce 100 simulations for each year (for each time slice and scenario), but due to the nature of volunteer distributed computing, not all model simulations are completed at the same time. Therefore, in this case, 200–400 simulations per year are sent out and whenever 100 simulations have been returned for each simulated year within a batch, this batch is closed and no additional simulation output is added to it. In cases when the minimum number of simulation per year did not reach 100 after some time, the batch was closed anyway, leading to a minimum number of simulations per year ranging from 85 to 100 depending on the scenario (Table 2).

Months 2-13 of the simulations being returned from each year are analysed, providing around 100 single-year simulations of data for each year (January to December), or a total of 10700 years of data for the historical baseline and 3000 years of data for each future time slice scenario.

For the historical baseline, the simulations are the same as those analysed by Guillod et al. (2016). SSTs and sea ice are prescribed to observed values using version 2 of the HadISST dataset (Rayner et al., 2003; Titchner and Rayner, 2014). Similarly, other input variables such as greenhouse gas concentrations, volcanoes and solar activity, $SO_2$ concentrations, etc, are prescribed to historical values as described in Guillod et al. (2016).

The future scenarios are 30-year time slices that correspond to years 1975–2004 of the historical baseline but with added climate change. Therefore, natural forcings (volcano and solar activity) are taken from 1975–2004, while greenhouse gases are taken from RCP8.5 for the simulated years (2020–2049 and 2070–2099). For sea surface temperature and sea ice, a similar approach is taken as in attribution studies (e.g., Schaller et al., 2016), but the future (rather than past) SST warming is added to (rather than subtracted from) observations. More specifically, the climate change signal derived from CMIP5 models (i.e., SST warming and corresponding changes in sea ice) is added to the 1975–2004 observed values used in the historical baseline. The details on the creation of future SST and sea ice is given in Sect. 3.2.



A number of daily and monthly variables are saved in the regional model (Table 3). Of particular relevance to hydro-meteorology and extremes, the following variables are available at daily time steps from the regional model output: minimum and maximum temperature (tasmin and tasmax, respectively), precipitation, surface air humidity (mean dewpoint temperature), mean sea level pressure, and additional variables required to compute potential evaporation ($E_{pot}$) (10m wind speed, and

incoming and net longwave and shortwave radiation fluxes at the land surface) as well as offline-computed $E_{pot}$ estimates (see Sect. 3.3 for details on the computation). In addition, 5-days averages of soil moisture on the four model levels as well as surface latent and sensible heat fluxes are available. All these variables, plus cloud cover and individual components of precipitation (convective versus large-scale, and snowfall versus rainfall) are available as monthly averages. Finally, weather@home is based on a calendar containing 360 days per year (i.e., 30 days per month), like many GCMs.

## 3.2 Sea surface temperature projections

To create the future SSTs and sea ice concentrations (SIC), two datasets are used: every available CMIP5 model simulation (Taylor et al., 2012), including all physics parameter and initial condition perturbations, and the HadISST2 observed SST and SIC (Rayner et al., 2003; Titchner and Rayner, 2014). The CMIP5 model data are used to produce the large scale warming patterns of SST for the two future time slices (2020–2049 and 2070–2099), whereas the HadISST data are used to provide the

small scale variability of the SST (whereby "small scale" refers here to anomalies from 30-year averages).

For the AMIP (climate model simulations with prescribed SSTs) component of the CMIP5 project, the projected change in SST and SIC are obtained from a single (per modelling group) coupled ocean / atmosphere model, and the models are integrated for a single decade from 2026–2035. This approach has two disadvantages. Firstly, using a single model does not take into account the variation in the ensemble of CMIP5 models, both in the global mean SST (GMSST) and the pattern of

warming produced. Secondly, the small scale variability of the SST patterns do not match those in our observed dataset, which makes comparison between the historical scenario and the two future scenarios difficult. To get around these problems we construct a statistical model of SST warming patterns and impose the small scale variability from the observed dataset, so as to match the historical scenario.

To construct the statistical model we use the SSTs for every model with data available for the RCP8.5 scenario. The below

analysis is carried out for each month in the datasets, so as to reflect the greater warming in the December-February season (DJF) in the CMIP5 ensemble. Firstly the SSTs are converted to anomalies by subtracting the 1986–2005 mean obtained from the corresponding historical run with the same model, run, initialisation and perturbation number. This gives a time series of SST anomalies for each CMIP5 ensemble member from 2006–2100. Secondly, to remove the small scale variability and generate the large scale warming patterns, a 30 year running–gradient filter is applied to every gridbox in the SST anomalies.

The statistical model of SST warming patterns is constructed from these smoothed SST anomalies by first performing an empirical orthogonal function (EOF, Wilks, 2011) analysis on the smoothed SST anomalies for the year 2050. This produces a set of patterns (the EOFs) and principal components (PCs) which explain the variation in the smoothed SST anomalies across the CMIP5 ensemble members. The number of EOFs and PCs was truncated at 6 as, during the analysis, it was determined that the first 6 accounted for 98% of the variability. As we are interested in producing transient series of SSTs for two periods,





the 6 EOFs were projected onto the smoothed SST anomalies for each year between 2020–2049 and 2070–2099 to produce timeseries of pseudo–PCs for each model and each year in the two scenarios. Next, a linear regression was performed on each set of pseudo–PCs for each year, to derive a relationship between the pseudo–PCs in that year and the PCs in 2050. These PC–relationships are used in the reconstruction of the SSTs later.

The core of the statistical model is a multi–variate distribution (MVD) of the truncated PCs in the year 2050, modelled by a Gaussian copula (Nelsen, 2007) with skew–normal marginals (Azzalini, 2005) using the "copula" and "sn" packages in the R statistical analysis software (R Core Team, 2016). A MVD is used as, although the EOFs are orthogonal to each other, the signs of the PCs within an ensemble member are not independent. Once the copula has been constructed it is sampled 10,000 times, which produces a set of 6 PCs for each sample. The SST warming pattern is then reconstructed from these PCs and

the EOFs for the year 2050, and the GMSST of the warming pattern is calculated and recorded with the PCs. This allows the construction of a probabilistic distribution of the GMSST warming in the CMIP5 ensemble which also contains the information (PCs) of how to construct the GMSST. Note that, for a given percentile, there will be 100 different sets of PCs. This allows the construction of up to 100 different warming patterns for each GMSST value, where the contributions to the mean warming occur in different physical locations. For this experiment we choose the 10th, 50th and 90th percentile values of GMSST so as

to incorporate CMIP5 models with both low and high sensitivity in their GMSST response to elevated GHG concentrations.

    Weather in the UK is potentially sensitive to the North Atlantic (NA) SSTs and in particular to gradients thereof (e.g., Rodwell et al., 1999; Rodwell and Folland, 2002). To account for this we use a NA SST gradient index to select the two most different warming patterns, in relation to this metric, from the 100 potential warming patterns for each of the 10th and 90th percentiles. This gradient is defined as the difference between the area weighted means of two areas in the North Atlantic,

following Schaller et al. (2016): A Southern area bounded by the longitude–latitude coordinates 30–50°N, 40–0°W and a Northern area bounded by 50–70°N, 40–0°W.

    From the sampling of the output of the copula we form 5 warming patterns for the year 2050, by combining the PCs with the EOFs: p10n corresponds to the pattern with a GMSST warming at the 10th percentile and the minimum NA SST gradient, p10x the 10th percentile GMSST and the maximum NA SST gradient, p90n the 90th percentile and the minimum NA SST

gradient and p90x the 90th percentile and maximum NA SST gradient, and MMM a median scenario with the median GMSST and middle NA SST gradient. Each of these patterns has an associated set of PCs for the year 2050. To generate a time series of SST anomalies the linear relationship between the original PCs in the year 2050 and the pseudo–PCs is used to construct time series of PCs for each of the 5 warming patterns above. These PCs (derived from the linear relationship) are then combined with the EOFs for the year 2050 to generate a time series of SST anomalies between the years 2020–2049 and 2070–2099 for

each of the 5 warming patterns.

    To generate absolute climatological SST values, the time series of SST anomalies are added to the 1986–2005 mean of the HadISST 2 dataset (since the above procedure was applied to anomalies from those same years). Since the future time slices are to be compared to the reference time period 1975–2004 (baseline), the small scale variability from these years is then also added onto the sum of the SST anomalies and HadISST mean. This small scale variability is also derived from the

HadISST 2 data by applying the 30 year smoother and then subtracting the smoothed data from the original HadISST 2 data.



This calculates the residuals of the smoother for 1975–2004 when compared to the original datasource and removes the large scale variability from HadISST, which was already added by the warming patterns.

To construct the sea–ice we use the 10 best CMIP5 models at representing historical sea–ice between 1979–2005, as ranked by Shu et al. (2015). For each future period (2020–2049 and 2070–2099), for every grid box we pool the SST anomalies for

the RCP8.5 scenario and the corresponding SIC anomalies. We then derive a linear relationship, for each grid box, between the SST anomaly and the SIC anomaly by using a linear regression. Timeseries of SIC absolute values are then constructed for each grid box by calculating the SIC anomaly from the timeseries of SST anomalies computed above and the linear relationship between SST anomaly and SIC anomaly. The 1986 to 2005 mean of the HadISST2 SIC is then added to the timeseries of SIC anomalies and then some post–processing is performed. Firstly, ice holes, which occur where a grid box with no ice is

surrounded by 8 grid boxes with ice, are filled with the mean value of the 8 surrounding grid boxes. Secondly isolated ice, where a grid box with ice is surrounded by grid boxes with no ice, are removed by setting the SIC in the grid box to 0. Thirdly, a longitudinal smoother is applied to the resulting data field.

As a result of this procedure, five SST time series are obtained for each future time slice (near and far future), which have the same small scale variability as the 1975–2004 HadISST SSTs and sample the inter-model variability in SST warming

from CMIP5 both in terms of GMSST and NA SST gradient. These 5 patterns are hereafter referred to as "scenarios" and are summarized in Table 2. Supplementary Figs. S1 and S2 display the resulting warming imposed on observed SSTs for near and far future scenarios, by season and scenario.

### 3.3 Potential evaporation estimates

Potential evaporation ($E_{\text{pot}}$) is defined as the amount of water that would evaporate from the land surface (soil, vegetation) into

the atmosphere if soil moisture supply was not limiting. Although a form of $E_{\text{pot}}$ is computed in the code of the land surface model MOSES 2, it cannot be directly saved as an output and must therefore be computed "offline" from the meteorological model output. Since $E_{\text{pot}}$ is an important variable that is used as an input to some impact models (e.g., hydrological models), this computation is done and the estimated $E_{\text{pot}}$ time series are included in the dataset along with the other variables. To do so, we estimate daily $E_{\text{pot}}$ (in $\text{mm day}^{-1}$) from the atmospheric model output based on the Penman-Monteith equation (Monteith,

1965) as follows (modified from Rudd and Kay, 2015):

$$E_{\text{pot}} = \frac{1}{\lambda} \frac{\Delta R_n + \rho_a c_a (e_s - e_d)/r_a}{\Delta + \gamma(1 + r_s/r_a)} \, , \tag{1}$$

where the following variables depend on the atmospheric variables: $\Delta$ is the rate of change of saturated vapour pressure with temperature ($\text{kPa}\,°\text{C}^{-1}$), $R_n$ is net radiation at the surface ($W\ m^{-2}$), $e_s$ is the saturation vapour pressure at near surface air temperature ($kPa$), $e_d$ is the near surface vapour pressure ($kPa$), $r_a$ is the aerodynamic resistance to vapour transfer in the

atmosphere ($\text{s m}^{-1}$) and $r_s$ is the bulk surface (canopy or bare soil) resistance ($\text{s m}^{-1}$). The following are constants in Eq. (1): $\lambda$ is the latent heat of evaporation ($2.45 \cdot 10^6\ \text{J kg}^{-1}$), $\rho_a$ is the near surface air density ($1\ \text{kg m}^3$), $c_a$ is the specific heat of air ($1013\ \text{J kg}^{-1}\,°\text{C}^{-1}$), $\gamma$ is the psychrometric constant ($0.066\ \text{kPa}\,°\text{C}^{-1}$).





The saturation vapour pressure $e_s$ can generally be computed from temperature $T$ (in °C) as

$$e_s(T) = 0.611 \exp\left(\frac{17.27T}{T + 237.3}\right) . \tag{2}$$

Therefore, we can derive

$$\Delta = \frac{de_s}{dT} = 17.27 \cdot 237.3 \frac{e_s(T)}{(T + 237.3)^2} \tag{3}$$

where $T$ is approximated by the average of daily minimum and daily maximum temperature ($T = (\text{tasmin} + \text{tasmax})/2$). For the computation of $e_s$ itself from daily data, however, we use a more accurate approach consisting of averaging $e_s$ values estimated from daily minimum and maximum temperature (tasmin and tasmax), i.e.,

$$e_s = \frac{es(\text{tasmin}) + es(\text{tasmax})}{2} . \tag{4}$$

The near surface vapour pressure can be directly estimated from daily averaged dewpoint temperature (tdps, in °C) based on
Eq. (2),

$$e_d = 0.611 \exp\left(\frac{17.27\text{tdps}}{\text{tdps} + 237.3}\right) , \tag{5}$$

and the aerodynamic resistance is computed from the daily mean 10m wind speed (wss, in $\text{m}\,\text{s}^{-1}$) using

$$r_a = \frac{243.489}{\text{wss}} , \tag{6}$$

hence including a logarithmic correction for wind height.

Finally, surface resistance $r_s$ is computed as in Rudd and Kay (2015), consistently with MORECS $E_{pot}$ estimates (Hough and Jones, 1997) and leading to the monthly surface resistance values shown in Table 4. $E$pot is not only affected by meteorological conditions, but also by vegetation. In particular, for future projections an important driver for vegetation is the ambient $CO_2$ concentration: plant stomata may need to open less widely with higher $CO_2$ concentrations, thereby conserving water (e.g., Keenan et al., 2013). Not accounting for this effect in offline $E_{pot}$ estimations has been shown to lead to an overestimation of
continental drying (Milly and Dunne, 2016), which is particularly relevant for drought analyses. Therefore, along with $E_{pot}$ estimates for future time slices using the same $r_s$ value as in the baseline (variable pepm), an additional variable (pepm_adjrs) is introduced which accounts for the impact of $CO_2$ on stomatal resistance and, therefore, on $E_{pot}$. To do so, we follow Rudd and Kay (2015) and use the estimate of change in crop and grass conductance per 1 ppm $CO_2$ concentration increase of Kruijt et al. (2008) ($-9.3 \cdot 10^{-2}\%$), and apply these to change in the 30-year averaged $CO_2$ concentration between each future time
period (469.5 ppm in the near future and 798.6 ppm in the far future) and 1975–2004 values (352.7 ppm). The resulting monthly $r_s$ values are displayed in Table 4 for pepm, and pepm_adjrs for near and far future.

Both variables (pepm and pepm_adjrs) are computed for each day using the $r_s$ value of the corresponding month, and monthly values are subsequently computed by averaging the daily values.





### 3.4 Generation of continuous time series from single years

Unlike other extreme events such as heat waves, heavy precipitation or cold spells, droughts often extend over months to years. While for short events (i.e., from a day to, say, one month), the direct use of single-year simulations can be suitable, longer, continuous time series are required to study droughts. However, a limitation of weather@home is that it can generate

simulations of only relatively short durations owing to the relatively slow computation on volunteers' personal computers. Here, we develop a methodology to derive plausible long continuous time series from a large ensemble of single-year simulations, whereby simulations in a given year are "stitched" to those of the next year using an appropriate criterion.

The criterion based on which simulations are "stitched" ideally ensures that the weather history of a simulation $S_y$ on year $y$, which is stitched to a simulation $S_{y+1}$ on year $y+1$, is consistent with the conditions found at the beginning of simulation $S_{y+1}$.

Given the slow nature of the temporal evolution of droughts, emphasis is put on obtaining continuous time series not necessarily from one day to the next, but rather at a temporal scale of the order of a week. Additionally, given the use of a large ensemble of simulations to construct multiple time series, the objective is not to derive time series that are really continuous (a task that may be considered impossible given the chaotic nature of the atmosphere), but rather to derive a set of time series that can be considered as continuous in the sense that their statistics can hardly be distinguished from those of continuous simulations.

Therefore, we focus on those components of the climate system that exhibit significant temporal memory (or auto-correlation) and that may impact the atmosphere. The ocean (i.e., sea surface temperature and sea ice) is a major component with these characteristics; however, being prescribed to observations in our simulations, it is continuous by definition and hence it does not need additional consideration for stitching purposes. Another such component is the land surface, in particular soil moisture. Soil moisture exhibits a few relevant characteristics: First, it exhibits memory of typically a few weeks to months (e.g., Koster

and Suarez, 2001) and, therefore, one may want to ensure that this memory is not lost in the stitching process – this may be particularly critical in the case of droughts. Second, the temporal evolution of soil moisture is mainly driven by precipitation minus evapotranspiration ($P - E$), i.e., by the weather in previous weeks to months. In other words, soil moisture can be seen as an approximate integrator of $P - E$ over time. Ensuring that soil moisture is continuous therefore also likely constrains the history of the weather, which in turn increases the temporal consistency in atmospheric conditions in the stitched time series

(for example, simulations with wet soils at the end of a year are likely to have exhibited wet conditions in December, while simulations with dry soils at the end of the year likely display less rainfall and higher temperature in December). Finally, soil moisture has been shown to be involved in key feedbacks relevant to droughts and heat waves (Seneviratne et al., 2010), such as soil moisture–temperature (Hirschi et al., 2011; Miralles et al., 2014) and soil moisture–precipitation (Roundy et al., 2013; Guillod et al., 2015) feedbacks. Therefore, ensuring continuous soil moisture avoids biases in the statistics of the weather in

the following few weeks. Note that this last characteristic is most relevant in transitional regions between wet and dry climate and is probably not critical in the UK in the winter season, when our simulations are stitched. Other variables that could have been considered include snow; however given that snow is not very frequent at the end of December over the UK, it may be difficult to distinguish between the large number of simulations which don't exhibit any snow at all.





Based on these considerations, we use soil moisture as a basis for stitching. Figure 1 displays an example of time series that are obtained with our simulation setup for two consecutive years, with the first month of the simulations (grey lines, implicitly part of a 13-months spin-up) leading to 12-months simulations (coloured). Stitching the 1990 simulations to the 1991 simulations is based on identifying the best match between the 1990 end-of-simulation values (last value for each simulation) and the value at the same time step in the 1991 simulations, i.e., the last value in the spin-up (in grey) leading to the 1991 simulations. Five-days averages (i.e., pentads) of soil moisture in the upper 1m of the soil (3 out of 4 model levels in this case) over the British Isles are used for this purpose.

While Fig. 1 is useful to understand the principle of the stitching methodology, the problem is more complex for gridded data as there are multiple locations (or grid cells), and, thus, multiple time series to consider for each set of simulations. An appropriate simplification of this problem is to ensure continuity of the main spatial patterns of soil moisture. To do this, we concentrate on the main modes of variability by computing the EOFs for the last pentad of December at the end of our simulations (Fig. 2). The leading EOF pattern is homogeneous in sign and thus characterises the overall soil moisture conditions within the analysed domain, while the second EOF characterises a Southeast–Northwest contrast. Together, these two leading EOFs explain 60% of the total variance, while further EOFs account for a much lower fraction of the variability (6% and lower). Hence we retain these two EOFs and use the reduced two-dimensional space of the principal components (PCs) corresponding to these EOFs (hereafter, PC1-2 space) to compare soil moisture fields and find similar conditions, defined by the lowest possible distance in this two-dimensional space.

The procedure used for stitching is the following:

1. Wait until a minimum of $n$ simulations is available for each year, which will allow the creation of $n$ time series (e.g., $n = 100$ for historical baseline).

2. Compute the PCs of soil moisture at the last pentad of December in months 1 ("start-of-run") and 13 ("end-of-run") of each simulation, i.e., obtaining the starting and ending soil moisture conditions.

3. Starting with the year $Y$ with the lowest number of simulations available ($= n$), all simulations are stitched forward as follows: the distance in the soil moisture PC1-2 space between each "end-of-run" value from simulations on year $Y$ and each "start-of-run" value from simulations on year $Y + 1$ is computed. The Hungarian algorithm (R function 'solve_LSAP' in package 'clue', Kuhn, 1955, 1956; Papadimitriou and Steiglitz, 1982; Hornik, 2005, 2016) is then applied to find the combination that minimises the sum of the squared distances.

4. The year $Y + 1$ simulations that have been selected are used and the previous step is repeated until the last year of the time series is reached.

5. The same procedure is applied backward, i.e., matching "start-of-run" values on year $Y$ to "end-of-run" values on year $Y - 1$. This is done repeatedly until the first year of the time series is reached.





The output of this procedure is a table which lists, for each time series, the simulation identifier for each year. The performance of the stitching methodology is evaluated from the historical baseline (1900-2006) by considering the soil moisture "error" obtained through stitching, using comparison of stitched and continuous simulations.

Fig. 3(a) shows the distribution of simulations in the PC1-2 space, for the last pentad in December. As detailed above, to create continuous time series, soil moisture at the end of the simulations on year $Y$ (month 13 of the simulation; last December pentad) is compared to soil moisture at the same time step in month 1 of simulations leading to year $Y + 1$. The distribution of the obtained distances in the PC1-2 space at the time of stitching is shown in black on Fig. 3(b). To evaluate this in the context of a continuous simulation, we analyse changes between consecutive soil moisture pentads in continuous simulation (continuous lines on Fig. 3b-d), taken from the last pentad in December to any of the first three pentads in January (i.e., transition at the beginning of our simulations). We find that the difference at the time of stitching (dashed black line) is substantially smaller than typical changes with a lag of one pentad in continuous simulations (continuous green line), both in terms of distance in PC1-2 space (Fig. 3b) and changes in these PCs considered individually (Fig. 3c,d), i.e., the soil moisture error is smaller than a temporal lag of one pentad. Furthermore, changes between the last December and first January pentads (i.e., with a lag of one) are only slightly larger in the stitched ensemble (dashed lines) than in continuous simulations (continuous lines). For a lag of three pentad (purple), the changes in soil moisture PCs are very similar in stitched and continuous simulations. In these panels, these changes can also be compared to what would happen in a randomly stitched ensemble (dotted lines). The changes in such an ensemble are, as expected, independent of the lag (since no temporal correlation is retained), and are substantially larger than those found in both the soil-moisture-stitched and continuous ensembles (dotted lines, lying on top of each other for all lags). These results show that the presented methodology allows to successfully stitch single-year simulations to each other, thereby ensuring consistency in weather statistics on time scales of weeks.

## 4   Validation of the historical baseline

The global and regional models in weather@home 2 have been validated thoroughly in Guillod et al. (2016) with respect to the simulated mean climate, trends and extremes, including the British Isles domain averages. Here, we further validate the 100 baseline time series at a more local scale over the UK. Section 4.1 investigates the biases in mean climate and describes the bias correction taken to alleviate major biases, while Sect. 4.2 focuses on hydro-meteorological extremes, i.e., low and high precipitation events.

### 4.1   Mean climate and bias correction

#### 4.1.1   Mean biases

Figure 4a–d shows the seasonal biases in surface air temperature with respect to the E-OBS dataset (Haylock et al., 2008). Biases are remarkably small for raw climate model output (within 1°C and often below 0.5°), with two main exceptions: a cold

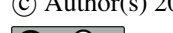



bias present in all seasons in the Northwest (Argyll region), and a warm bias in summer (June to August, JJA) in the South and Southeast.

Biases in precipitation with respect to the CEH-GEAR dataset (Fig. 4e–l), on the other hand, are more significant. In particular, precipitation is strongly underestimated in summer (20–50% or up to 1 $\mathrm{mm\,day^{-1}}$) and, to a lower extent, in autumn.

Conversely, winter precipitation tends to be overestimated in the Southeast. Possible mechanisms for these biases are discussed in Guillod et al. (2016). These biases have implications, particularly for the investigation of droughts and future drought risk, and the application of a bias-correction technique is therefore necessary. The next section (4.1.2) describes the approach chosen to correct precipitation data.

Another important variable for hydro-meteorological extremes is $E_{\mathrm{pot}}$, whose biases are shown in Fig. 4m–p with respect to

CHESS-PE (without interception correction) and highlight a small overestimation in summer (in the order of 20%). This bias is satisfactory, given large uncertainties in $E_{\mathrm{pot}}$ quantification, including the formula used.

### 4.1.2 Bias correction

Since biases in temperature and $E_{\mathrm{pot}}$ are relatively small, these two variables are not corrected. However, the substantial precipitation biases may be particularly problematic for drought analysis and correcting for these is therefore necessary. To do

so, a simple linear approach was chosen, using monthly bias correction factors (e.g., Lafon et al., 2013). The choice of bias-correction algorithm depends on the nature of the biases present and the uncertainty with which properties of the observed and modelled precipitation distributions can be estimated (e.g., Teutschbein and Seibert, 2012; Lafon et al., 2013). For example, if biases are present in higher order moments of the simulated precipitation distribution then more sophisticated bias correction techniques are warranted than if only the mean is biased. Nonetheless, the higher order moments of the precipitation record can

only be corrected if they can be estimated with confidence, which is not always possible for short-duration datasets. There is therefore a trade-off between reducing biases and introducing additional (often unconstrained) uncertainty. As recommended by Lafon et al. (2013), we use the simplest possible method which is able to correct significant biases in the data. In the present analysis we use a linear bias correction, which we calculate offers adequate correction of seasonal biases in the mean and which does not adversely affect higher order moments of the rainfall distribution. It is also noted that for drought studies using

climate model outputs the distribution of dry days (i.e., days with precipitation $< 1$ mm) can be important to preserve. In the present case we find that this distribution is maintained without further specific corrections (Supplementary Fig. S3). These were defined based on the overlapping time period between all observational datasets (CEH-GEAR, CHESS-PE, E-OBS) and our baseline, i.e., years 1961–2006. The mean precipitation for each calendar month was computed from the 100 baseline time series, and their ratio to the corresponding values in CEH-GEAR were computed (Supplementary Fig. S4). However, in

order to avoid sudden discontinuities between grid cells, a spatial smoothing was applied to the ratio using a 3-by-3 grid cells moving box and taking weights of 1/2 for the center box and 1/16 for the surrounding 8 boxes, leading to the precipitation bias correction factors shown in Fig. 5. Note that only the Great Britain coverage of CEH-GEAR data are used for bias correction, since CEH-GEAR data over Northern Ireland are available as a separate product and have not been processed.





Subsequently, daily and monthly precipitation values were multiplied by the factor for the corresponding month. The bias-corrected precipitation is also made available as part of the dataset as an additional variable (prbc, see Table 3). Unless explicitly mentioned, analyses in the rest of this study are based on bias-corrected precipitation data.

### 4.1.3 Inter-member variability

While the previous two subsections only consider the model climatology averaged from all 100 time series, part of the difference with observation may arise from natural variability, as expressed from the climatology of the individual time series. Indeed, although we often consider the observed climatology as the true climatology (albeit with some measuring errors), it is in fact one possible climatology among many and is determined by the one trajectory through the "weather phase space" that occurred by chance. This is due to the highly non-linear, chaotic behaviour of the atmosphere (e.g., Lorenz, 1965).

To assess the variability in climatologies in the 100 time series, Supplementary Figs. S5–S8 display the full range, interquartile range and median of climatologies (out of the 100 modelled climatologies) as well as observations for the 19 river basin regions used in UKCP09 (Murphy et al., 2009) within Great Britain. For temperature (Supplementary Fig. S5), all climatologies are relatively similar, but a larger spread is found for precipitation (Supplementary Figs. S6 and S7 for raw and bias-corrected values, respectively). Nonetheless, the observed climatology generally lies outside of all 100 climatologies for the main biases.

It should be noted that some biases persist after bias-correction (e.g., in the Western Highland and Tay regions on Fig. S7) due to the spatial smoothing applied to the bias correction factors. For $E_{\mathrm{pot}}$ (Supplementary Fig. S8), both variants from the CHESS-PE dataset are shown (with and without interception correction) and the main feature are captured relatively well, apart from an overestimation in the Southern regions in summer (see also Fig. 4).

### 4.2 Hydrometeorological extremes

In this section, the ability of the time series to represent the distribution of dry and wet extreme precipitation events is assessed, first at the scale of Great-Britain averages for prolonged dry periods (Sect. 4.2.1) and then at the regional scale for prolonged dry period and for shorter, high precipitation events (Sect. 4.2.2). Comparison to CEH-GEAR is done based on the overlapping years, i.e., 1900–2006, and using only data over Great Britain (CEH-GEAR data over Northern Ireland have not been processed).

### 4.2.1 Great Britain-averaged dry events

Fig. 6a–d show, for averaged values over Great Britain, return time plots of low precipitation (bias-corrected) cumulated over a whole season. For w@h, return values are displayed for each time series (grey) as well as when pooling all time series together (black). Overall, observed values lie within the range of the simulated values. However, w@h tends to overestimate winter low precipitation values (i.e., not dry enough) but underestimate summer low precipitation values (i.e., overestimated summer droughts).

Nonetheless, even in those cases there are individual time series which look similar to the CEH-GEAR dataset, suggesting that natural variability could explain some of those apparent biases.





While short droughts do not usually pose a serious threat to Great Britain, prolonged periods of drought (e.g., multi-annual) are more problematic. Therefore, we also show return time plots for multiple (one to four) consecutive hydrological years (October to September) on panels e–h of Fig. 6. At these longer time scales, the climate time series perform very well compared to the observed return values, which lie well within the ensemble. These results are encouraging for the MaRIUS project, as

they suggest that the dataset may well represent precipitation accumulation over long time period, which is the most critical aspect to British droughts. The next section goes into further details through validation at the regional scale.

### 4.2.2    Regional extremes

The analysis presented in the previous section was applied to regional averages of bias-corrected precipitation. To summarize the main findings, we focus on precipitation totals over multiple hydrological years and display, for each region, the distribution

of 100 return values estimated from the individual time series as boxplots, with the value estimated from the CEH-GEAR dataset overlaid as a white dot, for a number of return times (Fig. 7). Overall, the observed values lie well within the range of modelled values, with a few exceptions: in some regions (e.g., Northern Highland, North East Scotland, Northumbria, South East England and South West England) the time series slightly underestimate the values (i.e., overestimate drought intensity), while values are overestimated (i.e., dryness is underestimated) in the Western Highland region, probably due to the

remaining bias after correction (see Sect. 4.1.3). For shorter durations, a similar plot for low seasonal precipitation is shown in Supplementary Fig. S9 and allows dataset users to assess the performance of the dataset depending on their region and purpose.

Although the dataset was created within a project focusing on droughts, it could be used for other hydro-meteorological extremes such as floods. Therefore, we provide validation of high precipitation events at the regional level by focusing on total precipitation over a defined number N of consecutive days, rxNday for $N = 1$, 5 and 30 days. Figure 8 show the return

values for these 3 indices in a similar way as Fig. 7, but showing the results for both raw (uncorrected) and bias-corrected precipitation. The observed estimates are found to mostly lie within the spread of values obtained from the climate time series for raw precipitation, but less so for bias-corrected precipitation. This suggests that the model represents the processes related to high precipitation formation relatively well (e.g., representation of UK-scale dynamical systems and thermodynamic processes, Schaller et al., 2016), but has more difficulties to represent longer-term persistence – a common feature of climate

models. Therefore, we recommend the use of uncorrected precipitation values from the time series for studies that focus on high precipitation events.

It should be noted that the analysis of short-term events should be done on individual years separately rather than on the whole time series, e.g., rx5day should not lie at the transition from one year to the next since the weather is not strictly continuous (Sect. 3.4). For example, for rx5day, for each year, the first value is from January 1–5 and the last values from

December 25-30 (the 360 days in a year are split into 12 months of 30 days), but it may not be appropriate to use the five pentads that range from December 26–January 1 to December 30–January 4, in order to exclude undesirable concatenation of inconsistent weather systems.





## 5   Future projections

In this section, we display changes in the five far future scenarios with respect to the 1975–2004 baseline, while corresponding changes for the near future time slice are shown in the Supplement. First, changes in seasonal averages are displayed for the main variables, with a comparison to the UKCP09 projections (Murphy et al., 2009) at the regional level (Sect. 5.1). Indeed,

UKCP09 data provide, among others, projected changes for a number of climate variables, time periods and climate scenarios. Second, changes in extremes are investigated at the regional level for prolonged low precipitation period and for short, high precipitation extremes (Sect. 5.2).

### 5.1   Changes in mean climate

Fig. 9 shows the changes in mean temperature in all far future scenarios with respect to the baseline (1975–2004) and for each

season. Generally, temperature increases are highest in the scenarios with higher global mean SST increases (FF-p90x and FF-p90n) and lowest in the scenarios with low global mean SST increases (FF-p10n and FF-p10x). Consistently with UKCP09, temperature increases is largest in the Southeast and in summer in all scenario. Similar but lower increases in temperature are found in the near future time slice (Supplementary Fig. S10).

Fig. 10 shows the distribution of all possible changes in temperature (i.e., from all combinations of the future time series

with the baseline time series) and in UKCP09 (high emission scenario A1FI), relative to the years 1961–1990 for consistency with the UKCP09 data. The spread of UKCP09 values account for a wider range of uncertainty than in our time series, as it includes various climate models and parameter uncertainty. However, our various future scenarios generally cover the range of mean changes projected by the latest UK climate change scenarios.

The patterns of changes in seasonal mean precipitation (Figs. 11) highlight that, while in winter precipitation changes seem

mostly related to global mean SST increases (as for temperature), summer precipitation changes are most sensitive to the North Atlantic SST gradient: time series FF-p10n and FF-p90n induce the smallest precipitation decreases, while FF-p10x and FF-p90x lead to the largest precipitation decrease. Thus, large SST gradients in the North Atlantic (as defined by the metric described in Sect. 3.2) lead to drier summer conditions. It should be noted that changes in raw (without bias correction) precipitation are smaller in JJA, leading to an overall weaker drying in absolute terms. Similar patterns of change, but smaller

in amplitude, are identified in the near future time slices (Supplementary Fig. S11). By definition, relative changes are similar in both raw and bias-corrected precipitation as the same multiplicative factors are applied to both time periods.

Comparison of precipitation changes to UKCP09 (Fig. 12) reveals that the simulated time series lies on the dry end of the standard UK climate projections. The changes may thus be more similar to UKCP02, the previous UK climate scenarios, which were based on the same models that are used in w@h. This feature is important to keep in mind, especially when analysing

changes in drought. The dataset can thus be seen as an ideal test bed for dry conditions, but the actual future may potentially not be as dry as suggested by the climate time series presented in this paper.

Finally, projected changes in seasonal mean $E_{\text{pot}}$ are displayed in Fig. 13, using the $E_{\text{pot}}$ formulation where stomatal resistance is adjusted to $CO_2$ future concentrations. $E_{\text{pot}}$ substantially increase in summer, and to a lower extent in autumn





and spring. The changes are mostly driven by the global mean SST increase, similar to temperature and as one may expect due to the strong controls exerted by temperature on this variable. We note that not adjusting stomatal resistance to increased $CO_2$ concentrations in the future (Supplementary Fig. S12) would result in a significantly stronger increase in $E_{\mathrm{pot}}$, and therefore recommend to use pepm_adjrs for future analyses to prevent overestimating increases in drought. As for temperature and
precipitation, the near future time slice displays changes that are qualitatively similar to those of the far future but smaller in amplitude (Supplementary Figs. S13 and S14).

## 5.2   Changes in hydrometeorological extremes

As for the validation of extremes done in Sect. 4.2.2, we concentrate on extremes of low precipitation cumulated over a number of consecutive hydrological years, and on high precipitation extremes cumulated over a small number of consecutive years.

Fig. 14 displays the 10-year return value (i.e., 3rd highest value in each 30-year time series) of low precipitation accumulated over two hydrological years. The distribution of the values estimated from each time series is shown for the baseline and for each far future scenario. Generally, a strong drying is found, i.e., 10-year dry events are getting more intense. In most regions, most of the difference between the individual future scenarios (i.e., SST warming patterns) appears to be related to the North Atlantic SST pattern, rather than to global mean SSTs. This suggests, given the findings of Fig. 11, that the summer response
may drive the changes in longer droughts (2 hydrological years in this case).

Similarly, Fig. 15 displays the change in 10-year return value of rx5day, using uncorrected precipitation data since these perform better than bias-corrected for high precipitation events as highlighted in Sect. 4.2.2. High precipitation extremes are expected to increase in intensity in most scenarios, despite a smaller signal-to-noise ratio induced by the sampling of 10-year return values from 30-year time series. Unlike for drought, global mean SST increases appear to be the main factor leading to
the response in extreme high precipitation, consistently with the Clausius-Clapeyron relationship (higher SST leading to higher evaporation and higher moisture content) and with the current understanding of atmospheric thermodynamics (e.g., Schaller et al., 2016).

In the near future time slice, similar but smaller changes are found for low precipitation (Supplementary Figs. S15), i.e., a increase in drought severity may already be expected in this time period. However, for high precipitation events (rx5day,
Supplementary Fig. S16), the increase is very small in this time period and cannot be distinguished from natural variability.

## 6   Conclusions

This paper presents a new set of climate projections for the United Kingdom, based on a regional climate model driven by a global atmospheric model which accounts for uncertainty in the climate system response by sampling a range of changes in the ocean state from CMIP5 models. The dataset includes a large number of spatio-temporally consistent time series for the recent
past (1900–2006) and for the near and far future (30-yr time slices ending in the middle and at the end of the 21st century, respectively). Future projections follow the assumption of a high greenhouse gas emission scenario (RCP 8.5), allowing to test



the sensitivity of the system to relatively large changes in climate forcing. The analysis could be repeated for alternative RCP scenarios.

An advantage of this data set compared to previous UK climate projections is the availability of a large number of spatially consistent time series, which is important for risk analysis of hydrological phenomena that are sensitive to spatial and temporal
variability. This comes at the expense of essentially using only one climate model (global and regional). However, in an effort to sample as wide a range of conditions as possible, part of the uncertainty in the climate system response is incorporated by using a range of projected changes in ocean states from CMIP5 models.

One of the challenges associated with the chosen approach is the generation of continuous time series from a large set of single-year simulations. A novel methodology has been developed and validated, which is based on identifying simulations
with the best matching soil moisture patterns to ensure continuity in slowly-evolving hydro-meteorological variables, the ocean state being continuous by definition as it is prescribed. This methodology is shown to be a promising tool for the application of weather@home to long-lasting extreme events such as drought.

The created time series are shown to represent mean climate and extreme hydro-meteorological events relatively well, after correcting for a substantial precipitation bias. For high precipitation extremes, however, we find that the raw (uncorrected)
precipitation output performs better than bias-corrected precipitation; this highlights the need of an evaluation of the relevant metrics to chose the suitable set of variables to be used for studies using the climate data set, since the choice of bias correction depends on the intended application.

In the context of the MaRIUS project, these time series are being used as input to hydrological, ecological and agricultural models, among others. Combining these output with, for example, water resource models, will allow for an in-depth investi-
gation of the drivers of water scarcity in the UK and for the identification of suitable adaptation measures. Additionally, the availability of a large number of time series, driven by different SST patterns, will allow to identify the oceanic, meteorological, and hydrological drivers of drought in the UK in subsequent analyses. The spatio-temporal structure of drought in the UK, and how it may change in the future, will also be investigated as part of MaRIUS.

*Data availability.* The dataset will be made available on the Centre for Environmental Data Analysis (CEDA) platform after publication of
the paper in HESS discussion. Data, in NetCDF format, will be provided as yearly files for each simulation and with a table indicating the simulations corresponding to each time series and year, and will include the variables listed in Table 3 for all the time series for each scenario (Table 2).

*Author contributions.* B.P. Guillod designed the modelling experiments with input from R.G. Jones, N.R. Massey, J.W. Hall and M.R. Allen.
N.R. Massey created the future SSTs and sea ice boundary conditions. A.L. Kay assisted with the $E_{\mathrm{pot}}$ computation. B.P. Guillod, G.
Coxon, S.J. Dadson, R.G. Jones, G. Bussi and J. Freer designed and evaluated the bias correction methodology. S.N. Sparrow and D.C.H.
Wallom managed the climateprediction.net infrastructure and model simulations thereon. B.P. Guillod ran the model simulations, designed the stitching methodology, analysed the data and wrote the paper. All co-authors provided comments on the text.



*Competing interests.* The authors declare that they have no conflict of interest.

*Acknowledgements.* This work was undertaken within the MaRIUS project: Managing the Risks, Impacts and Uncertainties of droughts and water Scarcity, funded by the Natural Environment Research Council (NERC), and undertaken by a project team spanning the University of Oxford [NE/L010364/1], University of Bristol [NE/L010399/1], Cranfield University [NE/L010186/1], the Met Office, and the Centre for

5  Ecology and Hydrology [NE/L010208/1]. We acknowledge the E-OBS dataset from the EU-FP6 project ENSEMBLES (http://ensembles-eu. metoffice.com) and the data providers in the ECA&D project (http://www.ecad.eu). We also acknowledge the CEH-GEAR and the CHESS-PE datasets provided by the Centre for Ecology and Hydrology (https://eip.ceh.ac.uk). We are grateful to CEDA (Centre for Environmental Data Analysis, NERC) and their Jasmin analysis platform (Lawrence et al., 2013) on which data analysis has been done. We would like to thank our colleagues at the Oxford eResearch Centre: P. Uhe, A. Bowery and M. Rashid for their technical expertise. We would also

10  like to thank the Met Office Hadley Centre PRECIS team for their technical and scientific support for the development and application of weather@home. Finally, we would like to thank all of the volunteers who have donated their computing time to climateprediction.net and weather@home.





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



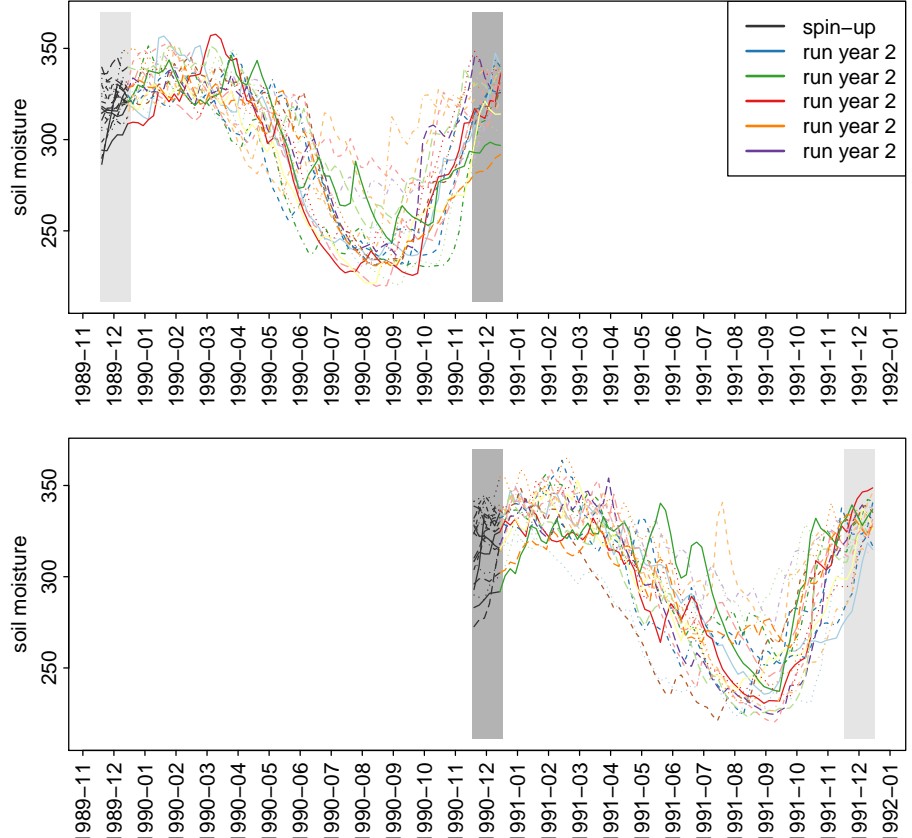

**Figure 1.** Illustration of the simulation design and stitching, using soil moisture model data averaged over the Thames catchment. Each panel shows, for a given year (top: 1990; bottom: 1991), 5-days averages of soil moisture in the upper 1 m of the soil for 20 model simulations. The first month of the simulations (December of the previous year, part of the spin-up) is indicated by grey lines, followed by the 12 months (January to December) in colours. End-of-year values of 1990 simulations and the same time steps in the spin-up leading to 1991 values, highlighted by dark grey boxes, have to be compared to find the combination allowing the best match between simulations. Light grey boxes indicates the same time steps that will be used to stitch to 1989 (top) and 1992 (bottom).





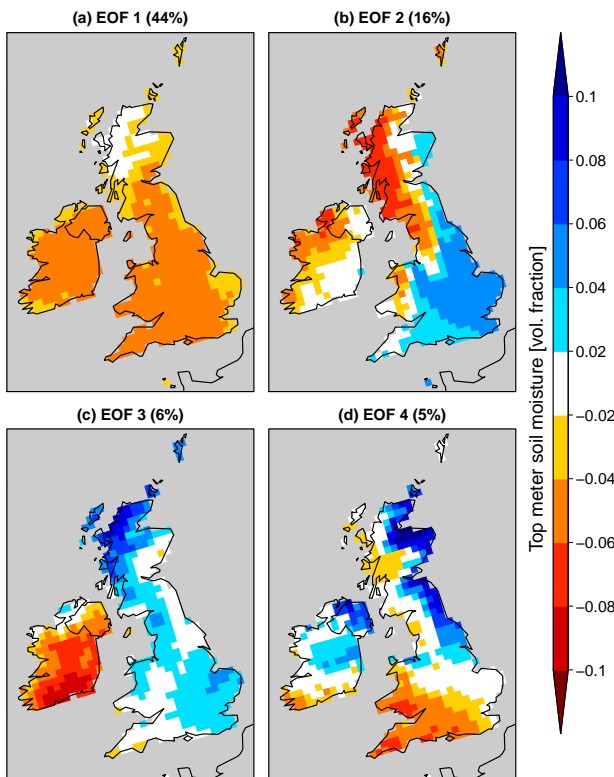

**Figure 2.** Leading EOFs of upper 1m soil moisture over the British Isles in the last pentad in December. The fraction of explained variance is indicated on each panel.




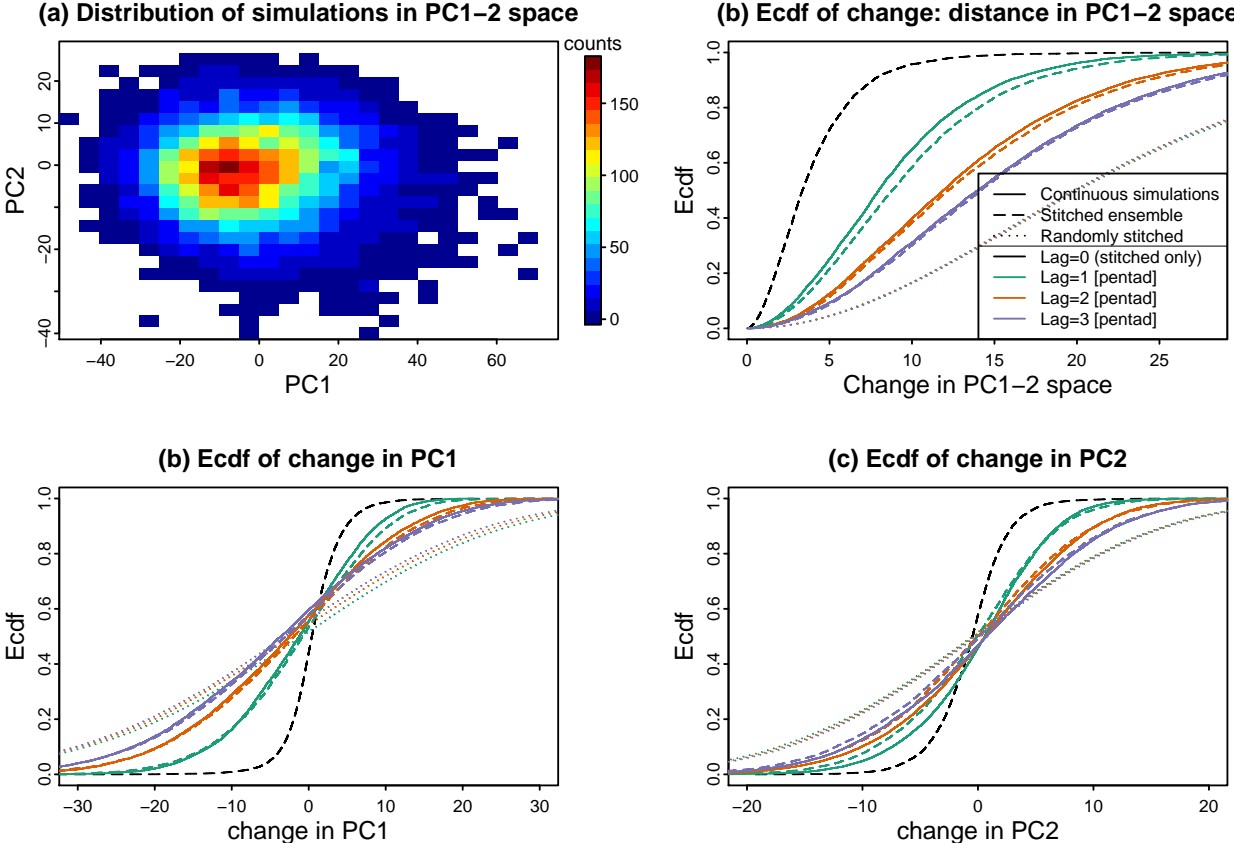

**Figure 3.** Temporal continuity of soil moisture in the stitched ensemble compared to continuous simulations in the historical baseline (1900–2006). (a) Bivariate distribution of soil moisture PCs 1 and 2 at the last December pentad. (b-d) Empirical cumulative distribution function of the changes in the PC1-2 space between the last pentad in December and the three subsequent pentads (colors) in continuous simulations (continuous lines), the stitched ensemble (thick lines) and a randomly stitched ensemble (points). The dashed black line shows the ECDF of the same distance at the time of stitching, i.e., between the same pentad in stitched simulations. (b) Absolute distance in the PC1-2 space, (c) Change in PC1 and (d) change in PC2.



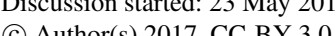

**Figure 4.** Seasonal biases in (a–d) mean surface air temperature, (e–h) precipitation in $\mathrm{mm\,day^{-1}}$, (i–l) precipitation in %, and (m–p) $E_{pot}$, for years 1961–2006. Each column is for a season as indicated in the labels.

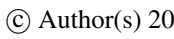


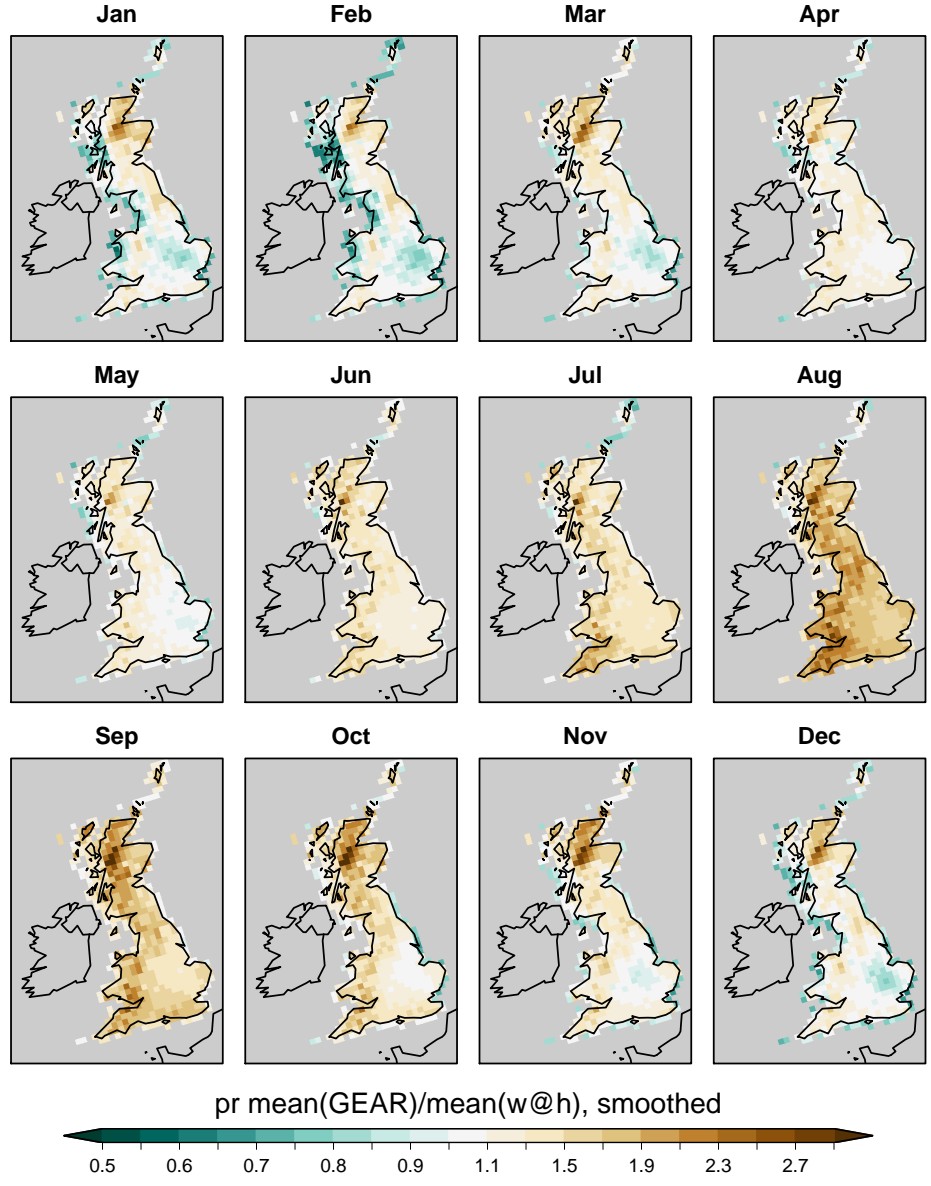

**Figure 5.** Bias-correction multiplicative factor applied to precipitation. A spatial smoothing was applied to the monthly ratios between observed (GEAR) and modelled (W@H) 1961–2006 precipitation (see Supplementary Fig. S4 for the unsmoothed ratio).





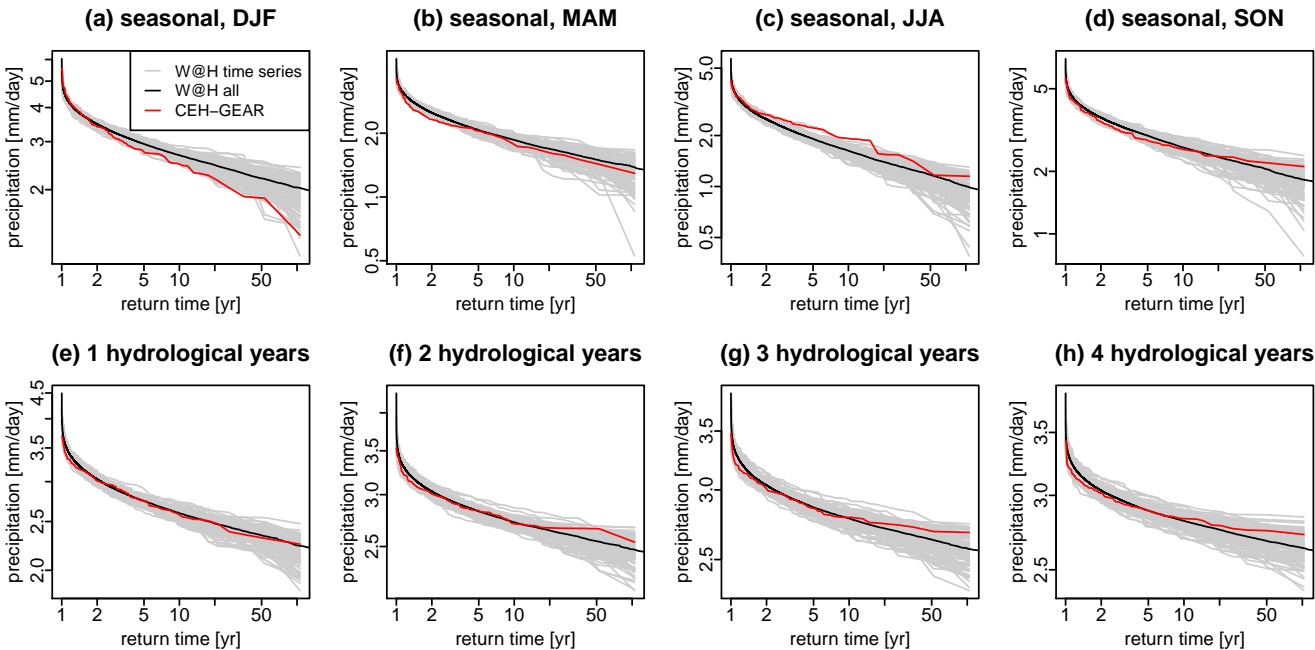

**Figure 6.** Return time plots of (a–d) low seasonal precipitation and (e–h) low precipitation accumulated in 1–4 consecutive hydrological years, for Great Britain averages from 1900–2006. (red) CEH-GEAR, (grey) individual w@h time series, (black) all w@h time series pooled together. For each time series, seasonal or (multi-)year averages of precipitation were computed and spatially aggregated over Great Britain prior to the computation of return values.





**Figure 7.** Return values of low precipitation accumulated over 1–4 hydrological years (x-axis) in the 100 baseline time series (boxplot) and in CEH-GEAR (white dot) for each region (panel), for return times of 5–50 years. See Supplementary Fig. S9 for the same analysis on seasonal precipitation rather than hydrological years. Whiskers display the range from individual time series.



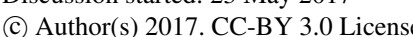

**Figure 8.** Return values of high precipitation indices rx1day, rx5day and rx30day (x-axis) in the 100 baseline time series (boxplot) and in CEH-GEAR (white dot) for each region (panel), for return times of 5–50 years (colour). Bias-corrected precipitation data is boxed in white (raw precipitation data in black). Whiskers display the range from individual time series. Note that for these metrics, the raw precipitation data compares better to observations than bias-corrected values.





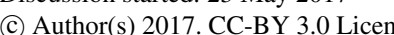

**Figure 9.** Difference in near-surface air temperature between far future and baseline (years 1975–2004 therein) for each season (row) and scenario (column). The corresponding figure for the near future time slices is shown in Supplementary Fig. S10.







**Figure 10.** Comparison of temperature projections with UKCP09: For each region, boxes show changes (2070–2099 minus 1961–1990) in JJA (left boxes) and DJF (right boxes) in the 5 sets of MaRIUS time series and in UKCP09 (high emission scenario: SRES A1FI; 10000 values available). Whiskers display the 10–90% range from each group.





precipitation (BC): far future minus baseline [mm day$^{-1}$]

**Figure 11.** Same as Fig. 9 but for precipitation (bias-corrected, prbc). The corresponding figure for the near future time slices is shown in Supplementary Fig. S11.





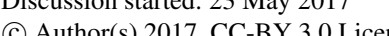

**Figure 12.** Same as Fig. 10 but for precipitation, in %.





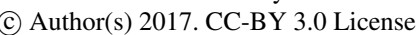

Figure 13. Same as Fig. 9 but for $E_{\text{pot}}$ (with stomatal resistance adjusted to $CO_2$ concentration; see Supplementary Fig. S12 for the changes when stomatal resistance is kept constant). The corresponding figures for the near future time slices are shown in Supplementary Fig. S13 and S14.





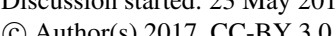

**Figure 14.** Distribution of return values of 10-year event for low precipitation on two consecutive hydrological years (boxplot) for each region (panel) and scenario (colour). Whiskers display the range from individual time series. The corresponding figure for the near future time slices is shown in Supplementary Fig. S15.





**Figure 15.** Distribution of return values of 10-year event for rx5day (boxplot) for each region (panel) and scenario (colour), using raw precipitation data (i.e., not bias-corrected). Whiskers display the range from individual time series. The corresponding figure for the near future time slices is shown in Supplementary Fig. S16.





**Table 1.** Observational datasets. For the mean climate validation the common overlapping period 1961–2006 is used, while for precipitation extremes validation the overlap period between the historical baseline and CEH-GEAR (1900–2006) is used.

| Variable | Dataset | Time period | Native resolution | Reference |
|---|---|---|---|---|
| Temperature | E-OBS (version 12.0) | 1950–2014 | 0.22° | Haylock et al. (2008) |
| Precipitation | CEH-GEAR | 1961–2014 | 1 km | Keller et al. (2015); Tanguy et al. (2015) |
| Potential evapotranspiration | CHESS-PE | 1961–2012 | 1 km | Robinson et al. (2015) |

**Table 2.** List of the climate time series for various scenarios. GM SST stand for Global Mean SST.

| Name | Short name | Years | GM SST percentile | SST North Atlantic index | # time series | Remark |
|---|---|---|---|---|---|---|
| Historical Baseline | bs | 1900–2006 | Observed (HadISST 2) | | 100 | "Baseline" refers to years 1975–2004 of the historical baseline |
| Near Future | nf | 2020–2049 | 50 | 50 | 100 | |
| Near Future p10n | nf-p10n | 2020–2049 | 10 | min | 91 | |
| Near Future p10x | nf-p10x | 2020–2049 | 10 | max | 91 | |
| Near Future p90n | nf-p90n | 2020–2049 | 90 | min | 89 | |
| Near Future p90x | nf-p90x | 2020–2049 | 90 | max | 85 | |
| Far Future | ff | 2070–2099 | 50 | 50 | 100 | |
| Far Future p10n | ff-p10n | 2070–2099 | 10 | min | 89 | |
| Far Future p10x | ff-p10x | 2070–2099 | 10 | max | 86 | |
| Far Future p90n | ff-p90n | 2070–2099 | 90 | min | 90 | |
| Far Future p90x | ff-p90x | 2070–2099 | 90 | max | 86 | |



**Table 3.** Output variables available in the dataset at various temporal frequencies.

| Temporal resolution | Variable name | Description | unit |
|---|---|---|---|
| daily only | tasmax | Maximum air temperature at 1.5 m above ground | K |
|  | tasmin | Minimum air temperature at 1.5 m above ground | K |
| daily & monthly | pr | Mean precipitation flux | mm s$^{-1}$ |
|  | prbc | Bias-corrected pr (Sect. 4.1.2) | mm s$^{-1}$ |
|  | pepm | Penman-Monteith potential evaporation (Sect. 3.3) | mm day$^{-1}$ |
|  | pepm_adjrs (future only) | Future Penman-Monteith potential evaporation with stomatal resistance adjusted to atmospheric $CO_2$ concentration (Sect. 3.3) | mm day$^{-1}$ |
|  | tdps | Mean dew point temperature at 1.5 m above ground | K |
|  | wss | Mean wind speed at 10 m above ground | m s$^{-1}$ |
|  | rsds | Mean incoming shortwave radiation at the surface | W m$^{-2}$ |
|  | rlds | Mean incoming longwave radiation at the surface | W m$^{-2}$ |
|  | rss | Mean net shortwave radiation at the surface | W m$^{-2}$ |
|  | rls | Mean net longwave radiation at the surface | W m$^{-2}$ |
|  | hfls | Mean latent heat flux at the surface | W m$^{-2}$ |
|  | psl | Mean sea level pressure | Pa |
| 5 days averages & monthly | hfss | Mean sensible heat flux at the surface | W m$^{-2}$ |
|  | moisture_content_of_soil_layer | Mean soil moisture content in each layer | m |
| monthly only | tas | Mean air temperature at 1.5 m above ground | K |
|  | prsn | Total snowfall flux | mm s$^{-1}$ |
|  | prrc | Convective rainfall flux | mm s$^{-1}$ |
|  | prsnc | Convective snowfall flux | mm s$^{-1}$ |
|  | clt | Fractional cloud cover | – |





**Table 4.** Monthly surface resistance values ($r_s$, in $\mathrm{s\,m^{-1}}$) used in the computation of $E_{\mathrm{pot}}$. The baseline values are shown under pepm, and are kept constant in future time slices for variable pepm. pepm_adjrs are future values accounting the changes in $CO_2$ concentration (see Sect. 3.3 for details).

| Months | pepm | pepm_adjrs (near future) | pepm_adjrs (far future) |
|---|---|---|---|
| January | 88.7 | 94.5 | 115.9 |
| February | 88.7 | 94.5 | 115.9 |
| March | 69.5 | 75.8 | 101.6 |
| April | 56.8 | 62.7 | 88.5 |
| May | 44.5 | 49.5 | 72.2 |
| June | 64.3 | 71.2 | 102.1 |
| July | 64.3 | 71.2 | 102.1 |
| August | 73.7 | 81.5 | 115.8 |
| September | 75.4 | 82.8 | 114.2 |
| October | 78.0 | 84.8 | 112.1 |
| November | 87.1 | 93.7 | 118.8 |
| December | 88.7 | 94.5 | 115.9 |