# Peer review of "A large set of potential past, present and future hydro-meteorological time series for the UK"

_Hydrology and Earth System Sciences, 2017_

## Referee Comment (RC1) · E. Blyth (Referee) · 21 Jun 2017

I think this paper describes a useful product that can be used by the scientific community. However I am a little concerned at the lack of analysis particularly of the bias in the model estimates of potential evaporation (PE). On page 12, lines 9 to 11, you state that there a a 'small overestimation' of the PE and claim this this is satisfactory. I think you might be brushing it off to casually. The bias is 20% which is not 'small'. Firstly, I think you should tell us where this bias is coming from. You mention earlier (Page 12, line 1) that there is a warm bias in the south in the summer. Presumably this is driving this 20% overestimate of PE. Secondly, I think you need to quantify what impact this will have on your drought estimates. Drought isn't only about rain - it's about drying as well. With a 20% overestimate of PE in the very dry area of the UK, you

might be overestimating the drought. Please add a paragraph or so on the drying bias. The many plots of results (figs 7, 8, 10, 12, 14 and 15) are hard to read. Firstly they all look much the same. Secondly, there is no map of where these regions are. You should have a map so that we know where catchments such as 'Dee' and Tweed' etc are. Not everyone has a geographical-hydrological map in their heads! In fact what I recommend (although it would take some time) would be to just do the 4 regions that we used in the CHESS paper (Robinson et al): Scotland, Wales, England and Lowland England. The advantage is that you have separated the climate zones of the UK and the plots are easier to digest as there are less of them.
* * *

---

## Referee Comment (RC2) · Anonymous Referee #2 · 10 Jul 2017

Dear Editor,

Comments to authors: Review of "A large set of potential past, present and future hydro-meteorological time series for the UK" (Manuscript number: hess-2017-246) submitted by BP. Guillod, RG. Jones, SJ. Dadson, G. Goxon, G. Bussi, J. Freer, AJ. Kay, NR. Massey, SN. Sparrow, DCH. Wallom, MR. Allen, JW Hall to Hydrology and Earth System Sciences (HESS; May 2017). This paper presents a new set of hydro-meteorological projections for the United Kingdom, based on a regional climate model driven by a global atmospheric model, which accounts for uncertainty in the climate system response by sampling a range of changes in the ocean state from CMIP5 models. This is really interesting papers, in particular as it describe a new methodology which could help in accounting better for internal climate variability, which is one of the

main source of uncertainty in the global/regional climate models, and which, at the local to regional scales, has been describe to be as important as anthropogenic climate change, even for intervals as long as the next 50 years in the middle and high latitude (Deser et al., 2012, 2013, 2014, 2016; Wallace et al. 2014, 2015), through the development of probabilistic scenarios for hydroclimate variables (including extremes such as drought) which could be used as input in a hydrological, ecological, agricultural models. It is therefore susceptible to interest a lot of researcher, and it is surely appropriated for publication in HESS. However, I have got some concerns regarding the absence of statistical analysis to test the significance of changes, and regarding the choice to only apply bias corrections to precipitation, while there is a clear significant bias in potential evapotranspiration (and probably in temperature). In addition, I feel like some results are a little over-interpreted, e.g. the raw (uncorrected) precipitation output performs better than bias corrected precipitation, as it can only be because the selected bias correction methods is not appropriated.

So my initial rank is to recommend major revisions.

Major comments: (1) Throughout the paper the authors are describing changes compared to a historical baseline, discrepancies between scenarios and while using different input (e.g. raw precipitation vs. bias-corrected precipitation), but none of these changes/differences are shown to be statistically significant. Although the novelty and the robustness of your approach in developing new hydro-meteorological projections is unquestionable, your interpretation of potential future changes in precipitation (including extreme events), potential evapotranspiration and temperature, or of discrepancies between scenarios, should still be supported be statistical tests (Student's t-test for changes in mean, F-test for changes in variance etc...) to make your results more objective or, at least, less subjective. In addition, as precipitation, temperature and evapotranspiration are likely to be auto-correlated, you might have to consider a test accounting for serial correlations in the time series.

(2) In Section 4.1.2 the authors decided not to apply a bias-correction to temperature
and evapotranspiration, as they consider their biases "relatively small". However, I don't think an overestimation of about 20% in evapotranspiration could be considered as small, and it could have a big impact on drought projections, as well as in hydrological and agricultural models which both used evapotranspiration as input. If this is really a small bias, it should be supported by a statistical test showing that bias is not significant, and that the results would have not been statistically different with or without applying bias correction. I agree there would always be large uncertainties when quantifying evapotranspiration, but, at least, this could be quantified (e.g. difference between different formulas, difference between raw and bias-corrected estimates), as it has been proposed in some studies (Sheffield et al., 2012; Zotarelli et al., 2013; Begueria et al. 2014; Raible et al., 2017). Overall, I would recommend to apply a bias correction to precipitation, temperature and evapotranspiration systematically, and then statically assess their potential differences.

(3) The choice of the bias correction is one of the main source of uncertainties while developing hydro-meteorological scenarios, and it different choices could lead to different results, more or less significant. The authors have chosen to use the simplest possible bias correction method, i.e. a linear bias correction, which is a fair choice knowing the large uncertainties related to the different bias correction procedures. However, at the same time, I would not recommend to conclude that the raw (uncorrected) precipitation output performs better than bias corrected precipitation, as another more sophisticated methods could have perform better (cf. Maraun et al., 2015; Maraun, 2016), and those uncertainties could have major implications, in developing scenarios for droughts or water resources (Clark et al., 2016). For instance, using a simple linear bias-correction, you should perform quite well in fixing the bias in the mean state, and probably the seasonal cycle (as it will captured most of the variance), but it might be totally unlikely to reduce the bias in interannual and/or decadal variability (while it is exactly where the climate models show the lowest skills; cf. Ault et al., 2012; 2013). This could explain why the bias looks proportionally stronger with a prolonged period of droughts in Figure 6. For instance, in a very recent study, Massei et al. (2017) proposed another

approach which could account better for low-frequency variability. However, I'm not sure this could applied in your study, and you must just be careful when interpreting your results, and should discussed more this issue.

Minor comments: P12-line 3-8: what would have been the bias if you would have used E-OBS for precipitation, and would have been the bias using E-OBS? Why did you not choose to keep the same data set for precipitation and temperature? P13-lines 29-30: It's especially true for the long return period (prolonged periods of droughts), do you have an explanation for that? It could be because your bias correction is only performant for the mean state, and for interannual variability which would expressed most of the variance, while lower-frequency variability might be important for long return period.

P-14-lines 3-6: Return time plots of low precipitation amounts in 1 –4 consecutive hydrological years primarily showing you that as much you increased the length of the records, as more the model shift from observations, in particular for return period greater than 10 years. However, this should more accurately tested by considering more consecutive days, and then comparing the results.

P-15-lines 20-22: summer precipitation changes are most sensitive to the North Atlantic SST gradient, but is the North Atlantic SST gradient likely to increase in summer in the coming 50 to 100 years? It would be great to discuss it in the paper (even briefly). In addition, the North Atlantic SST gradient is closely related to atmospheric zonal circulation patterns, such as the North Atlantic Oscillation (NAO). It would therefore be interesting to discuss the potential implications if the summer NAO was becoming more positively/negatively persistent in the future. It could be an interesting discussion.

P16 lines 14-15: This suggests a change in the annual cycle, which would be more sensitive to changes in the North Atlantic ocean-atmosphere coupled variability. I should be discuss more in conclusion, for instance.

[Figure]

246, 2017.

---

## Author Comment (AC1) · 7 Sep 2017

**Reply to the review by E. Blyth**

We are very grateful to E. Blyth for her review. We include our answers to the comments in blue font right under the unmodified comments from the review.

[Figure]

**REVIEW**

I think this paper describes a useful product that can be used by the scientific community. However I am a little concerned at the lack of analysis particularly of the bias in the model estimates of potential evaporation (PE). On page 12, lines 9 to 11, you state that there a "small overestimation" of the PE and claim this this is satisfactory. I think you might be brushing it off to casually. The bias is 20% which is not "small". Firstly, I think you should tell us where this bias is coming from. You mention earlier (Page 12, line 1) that there is a warm bias in the south in the summer. Presumably this is driving this 20% overestimate of PE. Secondly, I think you need to quantify what impact this will have on your drought estimates. Drought isn't only about rain - it's about drying as well. With a 20% overestimate of PE in the very dry area of the UK, you might be overestimating the drought. Please add a paragraph or so on the drying bias.

Thank you for this comment, which was also raised by referee #2. We agree with the referee that more discussion is needed on the PE bias. The PE bias can have various sources, including (but not limited to) the positive summer temperature bias. Other possible causes for the PE bias can be through its radiative (e.g., an overestimation of the net radiation through an underestimation of cloud cover) and aerodynamic (wind) components. These two additional possible sources of biases are difficult to correct owing to a lack of observations. Moreover, the uncertainty of the true PE is very large, as highlighted by the discrepancies from observational datasets (i.e. from datasets other than CHESS-PE).

To address this point, we have rewritten the discussion around PE biases in the manuscript. We have explained what the pitfalls of bias-correcting PE are in Section 4.1.2, justifying our decision not to correct these biases. We have also clearly stated at several places that the users of this dataset should be aware of this bias and,
depending on the application, should account for it in their analysis.

The many plots of results (figs 7, 8, 10, 12, 14 and 15) are hard to read. Firstly they all look much the same. Secondly, there is no map of where these regions are. You should have a map so that we know where catchments such as 'Dee' and 'Tweed' etc are. Not everyone has a geographical-hydrological map in their heads! In fact what I recommend (although it would take some time) would be to just do the 4 regions that we used in the CHESS paper (Robinson et al): Scotland, Wales, England and Lowland England. The advantage is that you have separated the climate zones of the UK and the plots are easier to digest as there are less of them.

This is a good point and indeed a map of the regions is lacking. We agree that a lower number of regions may be preferable, although we prefer to stick to the UKCP09 water regions as these are river basins. We have selected six representative regions (Western Highlands, North East Scotland, Tweed, North West England, Anglian and Thames) and have added a map of these regions (new Figure 7). The plots for all 19 regions are still included for interested readers, but as part of the supplement rather than in the main part of the paper (see also Supplementary Fig. S3 for a map of all regions).
* * *

---

## Author Comment (AC2) · 7 Sep 2017

**Reply to the review by Anonymous Referee #2**

We thank anonymous referee #2 for the positive review, which suggests useful additions to the manuscripts. These suggestions are highly appreciated. Answers to the comments are included in blue font right under the unmodified comments from the review.

**REVIEW** Dear Editor,

[Figure]

Comments to authors: Review of "A large set of potential past, present and future hydro-meteorological time series for the UK ' (Manuscript number: hess-2017-246) submitted by BP. Guillod, RG. Jones, SJ. Dadson, G. Goxon, G. Bussi, J. Freer, AJ. Kay, NR. Massey, SN. Sparrow, DCH. Wallom, MR. Allen, JW Hall to Hydrology and Earth System Sciences (HESS; May 2017). This paper presents a new set of hydro-meteorological projections for the United Kingdom, based on a regional climate model driven by a global atmospheric model, which accounts for uncertainty in the climate system response by sampling a range of changes in the ocean state from CMIP5 models. This is really interesting papers, in particular as it describe a new methodology which could help in accounting better for internal climate variability, which is one of the main source of uncertainty in the global/regional climate models, and which, at the local to regional scales, has been describe to be as important as anthropogenic climate change, even for intervals as long as the next 50 years in the middle and high latitude (Deser et al., 2012, 2013, 2014, 2016; Wallace et al. 2014, 2015), through the development of probabilistic scenarios for hydroclimate variables (including extremes such as drought) which could be used as input in a hydrological, ecological, agricultural models. It is therefore susceptible to interest a lot of researcher, and it is surely appropriated for publication in HESS. However, I have got some concerns regarding the absence of statistical analysis to test the significance of changes, and regarding the choice to only apply bias corrections to precipitation, while there is a clear significant bias in potential evapotranspiration (and probably in temperature). In addition, I feel like some results are a little over-interpreted, e.g. the raw (uncorrected) precipitation output performs better than bias corrected precipitation, as it can only be because the selected bias correction methods is not appropriated.

So my initial rank is to recommend major revisions.

We appreciate the overall positive tone of the reviewer's comments as well as the relevant points raised, for which we mention our intentions for the revised manuscript.

**Major comments**

[Figure]

1. Throughout the paper the authors are describing changes compared to a historical baseline, discrepancies between scenarios and while using different input (e.g. raw precipitation vs. bias-corrected precipitation), but none of these changes/differences are shown to be statistically significant. Although the novelty and the robustness of your approach in developing new hydro-meteorological projections is unquestionable, your interpretation of potential future changes in precipitation (including extreme events), potential evapotranspiration and temperature, or of discrepancies between scenarios, should still be supported be statistical tests (Student's t-test for changes in mean, F-test for changes in variance etc...) to make your results more objective or, at least, less subjective. In addition, as precipitation, temperature and evapotranspiration are likely to be auto-correlated, you might have to consider a test accounting for serial correlations in the time series.

To address this point, we have the tested statistical significance of the changes as follows: for the maps of changes in seasonal mean temperature, precipitation and potential evaporation, a two-sided T-test based on values from individual time series was applied, and grid cells where changes are not significant at the 95% level are hatched (Figs. 10,12,14 and corresponding figures in the supplement). For the comparison of our changes to UKCP09 (Figs. 11 and 13), the same approach cannot be used as UKCP09 only provides changes in the variables, rather than two sets of samples for baseline and future time slices. Thus, for these two figures we decided to consider a change as significant if 0 lies outside of the 5–95% range and to draw the boxes in grey for non-significant changes. For figures 15 and 16, the same procedure as for the maps was used: a two-sided T-test was applied to determine the statistical significance between in the mean of future scenarios and the mean of the baseline. Boxes whose mean do not significantly differ from the baseline are drawn in grey. Finally, we have added small edits to the results section where needed to account include for the

statistical significance information. We believe that these changes allow us to interpret the projected changes in a more objective fashion.

2. In Section 4.1.2 the authors decided not to apply a bias-correction to temperature and evapotranspiration, as they consider their biases "relatively small". However, I don't think an overestimation of about 20% in evapotranspiration could be considered as small, and it could have a big impact on drought projections, as well as in hydrological and agricultural models which both used evapotranspiration as input. If this is really a small bias, it should be supported by a statistical test showing that bias is not significant, and that the results would have not been statistically different with or without applying bias correction. I agree there would always be large uncertainties when quantifying evapotranspiration, but, at least, this could be quantified (e.g. difference between different formulas, difference between raw and bias-corrected estimates), as it has been proposed in some studies (Sheffield et al., 2012; Zotarelli et al., 2013; Begueria et al. 2014; Raible et al., 2017). Overall, I would recommend to apply a bias correction to precipitation, temperature and evapotranspiration systematically, and then statically assess their potential differences.
   A similar point was raised by referee #1 and we recognise that some discussion is needed around the potential evaporation (Epot) biases (note that the variable is potential evaporation, not actual evapotranspiration). However, we would not like to bias-correct temperature and even less so Epot as part of our product, for the following reasons. First, the Epot bias may have various origins, including temperature biases but also radiation (e.g. owing to biases in cloud cover) and aerodynamic (wind) components. The latter two are difficult to correct, owing to the lack of long-term gridded observations at the desired resolution. Second, observational estimates of Epot from various sources can significantly differ, depending on the assumptions and datasets used as input, implying that a bias-corrected Epot would be highly dependent on the chosen source of observed

values. Third, the assumption that the same bias correction can be applied to future scenarios would be even more questionable for Epot than for precipitation because of the inter-dependence of variables used to compute Epot. Fourth, the Epot provided in our dataset is one possible formulation and set of parameters, and some data users will compute Epot themselves using the other variables provided depending on their need. Therefore, we think that keeping our current approach is appropriate, provided that we clarify these points in the paper and warn potential users about the Epot bias and possible implications. We have added a paragraph justifying our approach in Section 4.1.2 as well as the following sentence in the conclusion: "We did not bias-correct potential evaporation but we strongly recommend data users to carefully assess possible impacts of these biases on their results, particularly with respect to drought analysis in the southern part of the UK".

3. The choice of the bias correction is one of the main source of uncertainties while developing hydro-meteorological scenarios, and it different choices could lead to different results, more or less significant. The authors have chosen to use the simplest possible bias correction method, i.e. a linear bias correction, which is a fair choice knowing the large uncertainties related to the different bias correction procedures. However, at the same time, I would not recommend to conclude that the raw (uncorrected) precipitation output performs better than bias corrected precipitation, as another more sophisticated methods could have perform better (cf. Maraun et al., 2015; Maraun, 2016), and those uncertainties could have major implications, in developing scenarios for droughts or water resources (Clark et al., 2016). For instance, using a simple linear bias-correction, you should perform quite well in fixing the bias in the mean state, and probably the seasonal cycle (as it will captured most of the variance), but it might be totally unlikely to reduce the bias in interannual and/or decadal variability (while it is exactly where the climate models show the lowest skills; cf. Ault et al., 2012;

2013). This could explain why the bias looks proportionally stronger with a prolonged period of droughts in Figure 6. For instance, in a very recent study, Massei et al. (2017) proposed another approach which could account better for low-frequency variability. However, I'm not sure this could applied in your study, and you must just be careful when interpreting your results, and should discussed more this issue.

We agree with the referee that our statement that the raw (uncorrected) precipitation output performs better than bias-corrected precipitation was misleading. The reason behind this counter-intuitive finding may indeed lie in the choice of bias-correction methodology and we have corrected our statement in the conclusion from
"For high precipitation extremes, however, we find that the raw (uncorrected) precipitation output performs better than bias-corrected precipitation; this highlights the need of an evaluation of the relevant metrics to chose the suitable set of variables to be used for studies using the climate data set, since the choice of bias correction depends on the intended application"
to
"For high precipitation extremes, the better performance of raw (uncorrected) precipitation output (compared to bias-corrected precipitation) highlights that while the choice of a simple linear bias correction might be appropriate with respect to mean, seasonality, and perhaps accumulated totals over a few months, analysis of short-duration hydrometeorological extremes might require the application of a more sophisticated bias-correction methodology. In addition, the application of a bias-correction technique to climate model output cannot correct for interannual to decadal climate variability, which is known to be poorly captured in current state-of-the-art climate models (e.g., Ault et al., 2012). This issue could potentially lead to an underestimation of the risk of multi-decadal droughts (Ault et al., 2014). As with any model-based dataset, an evaluation of metrics relevant to

the processes investigated is recommended in order to chose a suitable set of variables and, where required, to apply a suitable bias-correction technique".
In addition, we have reformulated and slightly expanded our description of this point in Section 4.2.2, where we now recommend the application of another bias-correction technique for the study of short duration, high precipitation extremes.

**Minor comments**

- P12-line 3-8: what would have been the bias if you would have used E-OBS for precipitation, and would have been the bias using E-OBS? Why did you not choose to keep the same data set for precipitation and temperature?
We have chosen to use precipitation data from the CHESS-met dataset because it is a widely used product in the UK. However, we had also computed the precipitation bias relative to the E-OBS dataset and the biases look qualitatively similar using that dataset (not shown).

- P13-lines 29-30: It's especially true for the long return period (prolonged periods of droughts), do you have an explanation for that? It could be because your bias correction is only performant for the mean state, and for interannual variability which would expressed most of the variance, while lower-frequency variability might be important for long return period.
We do not have an exact explanation for this. One possibility for the overestimation of low summer precipitation deficit at high return times could be an over-active soil moisture feedback, whereby an initial drying leads to a strong further drying owing to too little evaporation. However one can only speculate on the mechanisms here. Note that it is unclear how the plot (Fig. 6a-d) relates to low-frequency variability, as the variable shown is only a 3-month accumulation (the return time referring to the frequency of the values, not to the length of accumulation). Hence the plots show that the biases are larger for rarer events, not for

longer accumulation times (which are shown on panels e-h of the same figure). Nonetheless, we have added the following sentence at the end of the paragraph: "The difficulty for climate models to represent low-frequency variability (Ault et al., 2012), an aspect that is by definition not improved by bias-correction, could also play a role here.".

- P-14-lines 3-6: Return time plots of low precipitation amounts in 1–4 consecutive hydrological years primarily showing you that as much you increased the length of the records, as more the model shift from observations, in particular for return period greater than 10 years. However, this should more accurately tested by considering more consecutive days, and then comparing the results.
  We agree that the discrepancy between model and observations are smallest for the 1-year accumulation. However, no clear further increase in bias is found for longer accumulation times (2–4 years). It is somewhat surprising that the model overestimate drought at these durations, while the opposite is usually found for long droughts. We add the following sentence there to reflect this feature: "Noteworthy is a small overestimation of dryness at rare frequencies for long accumulation times (two to four years), not present in the one-year accumulated values, which suggests that in this case the climate model overestimates long-term precipitation persistence, unlike what has been shown for longer accumulation times (Ault et al., 2012)".

- P-15-lines 20-22: summer precipitation changes are most sensitive to the North Atlantic SST gradient, but is the North Atlantic SST gradient likely to increase in summer in the coming 50 to 100 years? It would be great to discuss it in the paper (even briefly). In addition, the North Atlantic SST gradient is closely related to atmospheric zonal circulation patterns, such as the North Atlantic Oscillation (NAO). It would therefore be interesting to discuss the potential implications if the summer NAO was becoming more positively/negatively persistent in the future. It could be an interesting discussion.

We added two sentences as follows, to remind the reader of how the North Atlantic SSTs are projected to change in our ensembles and to mention the NAO: "Note that the median scenario ("FF", called MMM in this figure for Multi-Model Median) exhibits the CMIP5 median change in this feature, while the four other scenarios depict extreme cases in both direction and should hence be considered as sensitivity scenarios. The mechanisms through which SST influence precipitation may include the North Atlantic Oscillation (NAO), which has been shown to be influenced by SSTs in the Atlantic and to influence European weather (e.g., Woollings et al., 2015)."

We agree with the referee that the link between the North Atlantic SST gradient and NAO is interesting. However, the scope of this paper is foremost to present the climate time series, while a detailed analysis of the sensitivity of atmospheric circulation (including the NAO) to future SST changes would be a separate paper.

- P16 lines 14-15: This suggests a change in the annual cycle, which would be more sensitive to changes in the North Atlantic ocean-atmosphere coupled variability. I should be discuss more in conclusion, for instance.

  Thank you for this comment. It is true that, given that summer and winter changes in precipitation are different in the different future scenarios (i.e., SST patterns), the future seasonal cycle will depend on the SST pattern. More generally, since we realised that a short paragraph on the projected changes in climate was lacking in the conclusion, we have added such a paragraph and have included a short mention of this aspect therein.

**References**

Ault, T. R., Cole, J. E., and St. George, S.: The amplitude of decadal to multidecadal variability in precipitation simulated by state-of-the-art climate models, Geophys. Res. Lett., 39, doi: 10.1029/2012GL053424, http://dx.doi.org/10.1029/2012GL053424, l21705, 2012.

[Figure]

Ault, T. R., Cole, J. E., Overpeck, J. T., Pederson, G. T., and Meko, D. M.: Assessing the Risk of Persistent Drought Using Climate Model Simulations and Paleoclimate Data, J. Clim., 27, 7529–7549, doi:10.1175/JCLI-D-12-00282.1, https://doi.org/10.1175/JCLI-D-12-00282.1, 2014.

Woollings, T., Franzke, C., Hodson, D. L. R., Dong, B., Barnes, E. A., Raible, C. C., and Pinto, J. G.: Contrasting interannual and multidecadal NAO variability, Clim. Dyn., 45, 539–556, doi:10.1007/s00382-014-2237-y, http://dx.doi.org/10.1007/s00382-014-2237-y, 2015.

---

## Author Response (AR2)

**Author's response**

We thank the reviewers for their useful comments which have greatly improved the manuscript. We also thank the editor.

We have corrected a few typos and inconsistencies: (1) The panels in Fig. 3 were wrongly labelled (a,b,b,c) and are now correclty labelled (a,b,c,d), and (2) we now consistently use "w@h2" instead of "w@h" where this was the case in the text, figure legends and figure captions).

Finally, the dataset has now also been published online so we have updated the "data availability" section to reflect this and cited the dataset reference.

Nothing else has been changed in the manuscript.